# Audio Signal-Stimulated Multilayered HfO_x_/TiO_y_ Spiking Neuron Network for Neuromorphic Computing

**DOI:** 10.3390/nano14171412

**Published:** 2024-08-29

**Authors:** Shengbo Gao, Mingyuan Ma, Bin Liang, Yuan Du, Li Du, Kunji Chen

**Affiliations:** 1School of Physics, Nanjing University, Nanjing 210093, China; 211840084@smail.nju.edu.cn; 2Collaborative Innovation Center of Advanced Microstructures, Nanjing University, Nanjing 210093, China; mingyuan_ma@smail.nju.edu.cn (M.M.); yuandu@nju.edu.cn (Y.D.);; 3School of Electronic Science and Engineering, Nanjing University, Nanjing 210093, China; 4Jiangsu Provincial Key Laboratory of Photonic and Electronic Materials Sciences and Technology, Nanjing University, Nanjing 210093, China

**Keywords:** artificial neuron and synapse, resistive switching, memory switching

## Abstract

As the key hardware of a brain-like chip based on a spiking neuron network (SNN), memristor has attracted more attention due to its similarity with biological neurons and synapses to deal with the audio signal. However, designing stable artificial neurons and synapse devices with a controllable switching pathway to form a hardware network is a challenge. For the first time, we report that artificial neurons and synapses based on multilayered HfO_x_/TiO_y_ memristor crossbar arrays can be used for the SNN training of audio signals, which display the tunable threshold switching and memory switching characteristics. It is found that tunable volatile and nonvolatile switching from the multilayered HfO_x_/TiO_y_ memristor is induced by the size-controlled atomic oxygen vacancy pathway, which depends on the atomic sublayer in the multilayered structure. The successful emulation of the biological neuron’s integrate-and-fire function can be achieved through the utilization of the tunable threshold switching characteristic. Based on the stable performance of the multilayered HfO_x_/TiO_y_ neuron and synapse, we constructed a hardware SNN architecture for processing audio signals, which provides a base for the recognition of audio signals through the function of integration and firing. Our design of an atomic conductive pathway by using a multilayered TiO_y_/HfO_x_ memristor supplies a new method for the construction of an artificial neuron and synapse in the same matrix, which can reduce the cost of integration in an AI chip. The implementation of synaptic functionalities by the hardware of SNNs paves the way for novel neuromorphic computing paradigms in the AI era.

## 1. Introduction

With the rapid development of big models and Chat GPT, spiking neural networks (SNNs) have attracted great attention as they provide a more biologically realistic approach to dealing with massive and diverse information [1,2]. Their sparsity, event-driven behavior, and bio-plausible local learning rules make SNNs suitable for low-power, neuromorphic hardware, bridging the gap between neuroscience and machine learning. Specifically, SNNs show perfect adaptability in audio signal processing. By utilizing SNNs, it becomes possible to better mimic the workings of the human auditory system [3]. While SNNs have found utility in sound classification and voice activation detection [4,5,6], the principal obstacle remains the development of artificial neurons and synapses with controllable switching pathways to establish a hardware network. As the key hardware of SNNs, brain-inspired synapse and neuron devices based on a memristor can mimic the human brain more accurately [7,8,9,10]. It is of vital importance to construct a spiking neural network by using synapse and neuron devices like the human brain. Within the realm of artificial synapses and neurons, the focus has been on HfO_x_/TiO_y_ memristors due to their compatibility with CMOS technology and their ideal resistance switching properties [11,12,13,14,15,16,17,18,19,20,21,22,23]. The seamless integration of artificial neurons and synapses utilizing HfO_x/_TiO_y_ memristors with neuromorphic chips necessitates highly desirable tunable threshold and memory switching attributes to ensure optimal compatibility with established CMOS technology standards. Different from the single layer of HfO_x_/TiO_y_, the resistive switching conductive pathway of the multilayered HfO_x_/TiO_y_ memristor can be controlled by the atomic thickness of the sublayer using atomic layer deposition technology. Because the size of the atomic conductive pathway is determined by the concentrations of oxygen vacancies in the two kinds of sublayers, the thicker the sublayer, the higher the oxygen vacancy. Moreover, the barrier of the HfO_x_ and TiO_y_ sublayers can be modulated by tuning the dielectric permittivity of TiO_y_ and HfO_x_ [24]. Because the number of oxygen vacancies in the conductive pathway is controlled by the sublayer thickness of the HfO_x_/TiO_y_ memristor, compared with the single TiO_x_ layer memristor and the single HfO_y_ layer memristor reported by Gul F. and Zhang Y. et al., a multilayered HfO_x_/TiO_y_ memristor has an advantage in stable endurance characteristics [25,26]. As for the neuron-like characteristic, the multilayered HfO_x_/TiO_y_ memristor can form a thinner oxygen vacancy pathway than that of the single-layered one due to the thickness of the TiO_x_ sublayer, and the HfO_y_ sublayer can be tunable on the atomic scale. As for the synapse-like resistive switching, the uniformity of the oxygen vacancy conductive pathway can be improved by the HfO_x_/TiO_y_ memristor due to the directional role of the thicker sublayer. Although HfO_x_/TiO_y_-based memristors have made lots of progress in the emulation of synaptic behaviors, the construction of SNN hardware based on the multilayered HfO_x_/TiO_y_ memristor to process audio signals has not yet been realized, which hinders the application of the multilayered HfO_x_/TiO_y_ memristor in SNN hardware for neuromorphic computing in the AI period.

In this article, we report that artificial neurons and synapses based on multilayered HfO_x_/TiO_y_ memristor crossbar arrays can be used for the SNN training of audio signals, which display the tunable threshold switching and memory switching characteristics. It is found that tunable volatile and nonvolatile switching from the multilayered HfO_x_/TiO_y_ memristor is induced by the size-controlled atomic oxygen vacancy pathway, which depends on the atomic sublayer in the multilayered structure. The successful emulation of the integrate-and-fire functionality of a biological neuron can be achieved through the effective utilization of the tunable threshold switching characteristics. Based on the stable performance of the multilayered HfO_x_/TiO_y_ neuron and synapse, we designed a hardware SNN architecture for processing audio signals, which provides a base for the recognition of audio signals through the function of integration and firing. Our implementation of synaptic functionalities in the hardware of SNNs paves the way for novel neuromorphic computing paradigms in the AI era.

## 2. Materials and Methods

This study details the fabrication process of a system comprising a multilayered HfO_x_/TiO_y_ memristor crossbar array. Initially, the Si substrate underwent cleaning following a standard procedure. Utilizing the wet thermal oxidation technique, layers of SiO_2_, measuring 500 nm in thickness, were then deposited atop silicon wafers. Subsequently, standard photolithography methods were applied to put patterns onto the top layer of SiO_2_. Ti was electronically evaporated onto the former patterned SiO_2_ layer to build the bottom electrodes (BEs), with lift-off processes following. Thirdly, with the aid of an atomic layer deposition (ALD) system, the top layer of the BE was evenly covered by multilayered HfO_x_/TiO_y_ films with 6 periods. The thickness of the HfO_x_ and TiO_y_ layers was 2 nm and 0.5 nm, respectively. We made the layer-by-layer growth of HfO_x_ and TiO_y_ by using the OpAL system of the Oxford Instruments company. We used Hf[N(C_2_H_5_)CH_3_]_4_ (TEMAH) and Ti[N(CH_3_)_2_]_4_ (TDMAT) as the main precursors combined with H_2_O for the fabrication of multilayered HfO_x_/TiO_y_ films. During the deposition process, the chamber pressure was held at 4 mTorr and the temperature was 573 K. Hf[N(C_2_H_5_)CH_3_]_4_ (TEMAH) was pulsed into the chamber, which was purged by Ar gas. Then, H_2_O was pulsed into the chamber, which was also purged with Ar gas. After pumping, the Ti[N(CH_3_)_2_]_4_ was pulsed into the chamber, followed by purging with Ar gas. Then, H_2_O was pulsed into the chamber with Ar gas to complete the purging process. The HfO_x_ sublayer and TiO_y_ sublayer were deposited alternately for six periods to form the multilayered HfO_x_/TiO_y_ films as depicted in Figure 1d. Fourthly, a GSE etching system was used to remove the multilayered HfO_x_/TiO_y_ deposition on the surface of the Ti BEs. Ultimately, utilizing standard photolithography and lift-off processes, the surface of the multilayered HfO_x_/TiO_y_ films received the electronic beam evaporation of platinum; thus, top electrodes (TEs) were formed, measuring 200 µm in diameter. The thickness of the Ti and Pt electrodes is 40 nm. A PHI 5000 Versa Probe (Ulvac-Phi Inc., Chigasaki, Japan) was employed to perform XPS tests for the atomic concentration ratios of the multilayered HfO_x_/TiO_y_ films. With a JEOL 2100F electron microscope operated at 200 kV, high-resolution cross-section transmission electron microscopy (HRXTEM) (JEOL Inc., Tokyo, Japan) was carried out for the microstructure analysis of the multilayered HfO_x_/TiO_y_ memristor. To test the electrical properties of the multilayered HfO_x_/TiO_y_ memristor, an Agilent B1500A semiconductor analyzer (Agilent Inc., Santa Clara, CA, USA) was applied for measurement under atmospheric conditions.

## 3. Results and Discussion

Figure 1a illustrates the detailed structure of the multilayered HfO_x_/TiO_y_ memristor crossbar arrays. The multilayered HfO_x_/TiO_y_ films are sandwiched between the BE and the TE. Figure 1b displays the optical microscope image of the crossbar structure. The amplified image of the intersection point is displayed in Figure 1c. The microstructure of the HfO_x_/TiO_y_ multilayers is revealed in Figure 1d through the high-resolution transmission electron microscopy (HRTEM) image. The thickness of the HfO_x_ and TiO_y_ sublayers is 2 nm and 0.5 nm, respectively. The raw data and Gaussian peak fitting outcomes of the X-ray photoelectron spectroscopy (XPS) spectra for Hf 4f and O 1s in the HfO_x_ sublayer are displayed in Figure 1e,f. To eliminate charging effects, the calibration of all XPS spectra was conducted with reference to the C 1s peak at 284.8 eV. The XPS spectrum in Figure 1e reveals two peaks at 16.49 eV and 18.15 eV, corresponding to the Hf^4+^ 4f_7/2_ and Hf^4+^ 4f_5/2_ states, respectively. As shown in Figure 1f, the O 1s spectrum exhibits a single peak at 530.14 eV, indicating the presence of only Hf^4+^ and O^2-^ in the hafnium oxide film. It is further confirmed that the nature of the HfO_x_ sublayer is stoichiometric. Figure 1g,h illustrate the original XPS data and Gaussian peak fitting results for Ti_2p_ and O_1s_ in the TiO_x_ film, respectively. As shown in Figure 1g, the Ti^4+^ 2p_3/2_ and Ti^4+^ 2p_1/2_ peaks are fitted with two binding energy peaks at 458.64 eV and 464.49 eV, while the Ti^3+^ 2p_3/2_ and Ti^3+^ 2p_1/2_ peaks are fitted with two binding energy peaks at 458.02 eV and 463.56 eV, respectively [27,28,29,30,31]. This indicates the presence of both Ti^4+^ and a certain proportion of Ti^3+^ valence states. By calculating the fitted peak areas, the proportions of Ti^4+^ and Ti^3+^ are found to be 75.2% and 24.8%, respectively. In Figure 1h, the XPS spectrum can be deconvoluted into two peaks, including 530.1 eV and 531.7 eV [32,33], which correspond to the Ti-O ionic bond and the oxygen vacancies in the TiO_y_ sublayer. The proportions of O^2−^ and oxygen vacancies are 74.5% and 24.5% by integrating the fitted O peak areas, respectively. They are consistent with the valence state of Ti. Consequently, the TiO_y_ sublayer is not a standard stoichiometric oxide, containing a number of oxygen vacancies.

As depicted in Figure 2a–c, the shift from tunable threshold to memory switching was observed from the multilayered HfO_x_/TiO_y_ memristor device as the compliance current (Icc) increased from 1 µA to 1mA. When the positive sweeping voltage reaches a SET threshold voltage, as indicated by the yellow dashed line, the HfO_x_/TiO_y_ memristor transitions from the high-resistance state (HRS) to the low-resistance state (LRS) across a range of 1 µA to 100 µA in Icc. The corresponding Icc values of 1 µA, 10 µA, and 100 µA align with the SET threshold voltages of 2.2 V, 2.9 V, and 3.3 V, respectively. The increasing Icc results in a stronger electric field intensity, as indicated by the observation. Following this, the device reverts back to the high-resistance state (HRS) automatically when the sweeping voltage falls below the holding voltage, denoted by the yellow dashed line. It should be noted that as the Icc decreases, the holding voltage increases. The holding voltage values of 0.63 V, 0.51 V, and 0.33 V correspond to the current values of 1 µA, 10 µA, and 100 µA, respectively. When the Icc increases to 1 mA, the HfO_x_/TiO_y_ device exhibits a memory switching (MS) characteristic, as depicted in Figure 2d. Resistive switching from the high-resistance state (HRS) to the low-resistance state (LRS) is observed under a positive bias, with the LRS persisting for an extended duration after the voltage is removed, showcasing a characteristic feature of nonvolatile switching. A switch from the low-resistance state (LRS) to the high-resistance state (HRS) is detected under a negative bias, demonstrating a bipolar switching characteristic. The reproducible performance, with a memory window of 3 × 10^5^, can be observed following 100 cycles, as depicted in Figure 2e. The accumulative probability of the Vset/Vreset values in Figure 2f shows that the narrower distribution of device-to-device parameters in our device serves as additional confirmation of its remarkable uniformity. The stability of both the HRS and LRS in the multilayered HfO_x_/TiO_y_ memristor device can be maintained for 10^4^ s at a reading voltage of 0.01 V, showcasing the characteristic of good retention, as depicted in Figure 2g. When the Icc is decreased to 1 µA, the statistical likelihood of threshold switching approaches 100%. Upon increasing Icc to 10 µA, the statistical probability of threshold switching decreases to 65%, validating that the transition between threshold switching and memory switching can be achieved by adjusting the value of Icc, as confirmed by Figure 2h.

In order to elucidate the underlying mechanisms governing the transition between threshold switching and memory switching in the multilayered HfO_x_/TiO_y_ memristor with greater precision, we developed a schematic diagram model depicting the pathway of the device under varying values of the Icc. As shown in Figure 3a, the initial TiO_y_ sublayer contains numerous oxygen vacancies, resulting in a relatively low resistivity. In contrast, the HfO_x_ sublayer has a relatively high resistivity due to the standard stoichiometric ratio. With a lower Icc, the electric field strength is reduced, resulting in the accumulation of a small quantity of new oxygen vacancies in the HfO_x_ and TiO_y_ sublayers, leading to the formation of a fragile and thin conductive pathway. The device transitions from the HRS to the LRS due to the establishment of a delicate oxygen vacancy pathway. The resistance increases as the conductive pathway becomes thinner. The fusion of the oxygen vacancy pathway can occur as a result of the joule heating from the current flow [34]. And the continuous oxygen vacancy pathway near the bottom electrode will be broken. Consequently, the device can revert back to the HRS, revealing the reason for the occurrence of threshold switching in the multilayered HfO_x_/TiO_y_ device. Different from the lower Icc, a large number of oxygen vacancies can be produced in the TiO_y_ and HfO_x_ sublayers under a higher Icc. Oxygen atoms are dislodged from the crystal lattice under the higher Icc, transforming into freely mobile oxygen ions. These oxygen ions drift towards the electrode, leaving behind immobile oxygen vacancies. With an elevated Icc and a positive electric field, the magnitude of the electric field surpasses that of lower Icc scenarios. Consequently, an increased number of oxygen vacancies are generated as the positive voltage rises [35,36,37], leading to the transition of the device to the LRS. The abundance of oxygen vacancies contributes to the development of a more robust and thicker oxygen vacancy conductive pathway. When subjected to a negative electric field, the oxygen ions migrate back to recombine with the oxygen vacancies, causing the device to transition to the HRS, as depicted in Figure 3b. As a result, memory switching becomes observable in the multilayered HfO_x_/TiO_y_ device under elevated Icc circumstances. The conductive pathway model can be further revealed by the relation betweenthe Icc and Vset. As displayed in Figure 2a–d, the Vset corresponding to Vth increases from 2.2 to 3.6 V with the compliance current changing from 1µA to 1mA, respectively. When the Icc was restricted relatively low, the formation of a thinner oxygen vacancy conductive pathway was unstable and fragile. As the voltage swept back, the thinner conductive pathway was more easily broken, leading to a sudden decline in current, as shown in Figure 2a–c. The driving force for the formation of the thinner conductive pathway is the intensity of the electric field induced by the Vset. The stronger the electric field intensity, the thicker the conductive pathway. As the Icc is enhanced from 1 µA to 1 mA, the diameter of the conductive pathway becomes thicker, which needs a high Vset to be sustained. It is further demonstrated that the threshold characteristic of multilayered HfO_x_/TiO_y_ films is induced by the thinner oxygen vacancy conductive pathway, which is determined by the lower electric field intensity in the device. It is noteworthy that the process of transitioning from threshold switching behavior to memory switching state by increasing the compliance current (Icc) in the multilayered HfO_x_/TiO_y_ memristor is not reversible. This is because a fragile and thin conductive filament is formed at a lower Icc due to the accumulation of a small number of oxygen vacancies in the HfO_x_ and TiO_y_ sublayers. This fragile filament leads to the threshold switching behavior. When the Icc is increased substantially, a large number of oxygen vacancies are generated in both sublayers, resulting in the formation of a more robust and thicker conductive filament. This stable filament can make the device switch to the memory switching state, exhibiting bistable resistance switching. The conductive filament is stable and difficult to disrupt or revert back to its fragile state under lower Icc conditions.

Because of the consistent volatile switching characteristics exhibited by the multilayered HfO_x/_TiO_y_ device using a compliance current (Icc) of 1 µA, we employed the integrate-and-fire (IF) function through the application of consecutive identical pulses to the device, as demonstrated in Figure 4a. According to the pathway model, under the influence of successive positive pulses applied to the device, the oxygen ions will be compelled in the negative direction, leading to the gradual formation of the oxygen vacancy pathway, resembling the ion influx observed in biological neurons. Once the oxygen vacancy pathway establishes a robust connection with the electrodes, there is a sudden drop in resistance, resulting in a significant spike in the current. This signifies the successful transition of the device into the firing state. After the firing process, the oxygen vacancy channel narrows significantly. In the low-resistance state, the current passing through the oxygen vacancy channel generates joule heating, leading to channel fusing. This causes the resistance switching channel to become open-circuited, thereby resulting in the cessation of firing. As a result, the device reverts back to the HRS. Illustrated in Figure 4b, the device failed to trigger when the voltage pulse’s amplitude and width were set at 0.5 V and 100 µs, respectively, with a 200 µs interval. Nonetheless, the firing state became distinctly apparent following the elevation of the voltage amplitude to 1.0 V, as evidenced in Figure 4c. Upon further increasing the voltage amplitude to 1.5 V, the firing frequency escalated, as depicted in Figure 4d. Hence, the firing frequency can be affected by the amplitude of the applied pulse. It is interesting to find that the firing frequency increases obviously when the pulse width is reduced from 100 µs to 60 µs and 20 µs, with the voltage amplitude remaining at 1.5 V as shown in Figure 4d–f. Therefore, the firing frequency also depends on the applied pulse frequency.

In order to complete the SNN training based on the multilayered HfO_x_/TiO_y_ synapses and neurons, we designed the circuit diagram of SNNs according to the structure of the biological neuron. Figure 5a illustrates the schematic diagram of a biological neuron, which consists of the presynaptic terminal, neuron cell body, and postsynaptic terminal. The presynaptic terminal receives the input signal, which passes through the neuron for processing and finally arrives at the postsynaptic terminal. Based on the integration and firing characteristics of the multilayered HfO_x_/TiO_y_ neurons, we propose a hardware implementation of SNNs consisting of three pre-neurons and two post-neurons connected to each other through six synapses as shown in Figure 5b. It is worth noting that the HfO_x_/TiO_y_ memristor device can be used as a synapse and a neuron separately in the SNN under different compliance currents of Icc. As shown in Figure 5, the three artificial HfO_x_/TiO_y_ memristors named N1, N2, and N3 exhibit threshold switching behavior under lower Icc conditions. They can mimic the integrate-and-fire characteristics of a pre-neuron. In this mode, they are defined as the neuron component in the SNN. Similar to the functions of N1, N2, and N3, the two artificial HfO_x_/TiO_y_ memristors named N4 and N5 play the role of a post-neuron in the SNN under a lower Icc. Conversely, under high Icc conditions, the two artificial HfO_x_/TiO_y_ memristors named S1 and S6 display memory switching and emulate the weight update behavior of a synapse. In this mode, the HfO_x_/TiO_y_ memristor devices serve as the synapse component in the SNN. In this architecture, input pulses are first processed by the neurons at the network’s frontend. Subsequently, the processed signals pass through synapses, where they are weighted. Finally, the signals propagate to the backend neurons, which are integrated and fired. Based on the stable performance of the multilayered HfO_x_/TiO_y_ memristor device, we have developed a hardware pulse neural network model triggered by audio signals, as illustrated in Figure 5c. Here, the audio signals are initially sampled by a microphone. After pre-processing and one-hot encoding, these signals are transformed into pulse signals, which are fed into the SNN hardware. The pulse signals are integrated and fired to give output signals.

To demonstrate the training process of our designed SNN for audio signals, we conducted the following performance by introducing pulses named signal1 and signal2 into channels named input1 and input2 in different sequences. Note that the synaptic weight of synapse S1 is larger than that of the synapse S2. Here, the two pulses arrive at the pre-neuron in different sequences with an interval of 100 μs. We employed cosine pulses as signal1 and signal2 and exchanged their arriving sequences for input1 and input2 as shown in Figure 5d,g. The overlapped input is depicted in Figure 5e,h. According to Kirchhoff’s law, the input is the summate current passing through S1 and S2. When signal1 arrives at S1 with a greater weight before signal2 arrives at S2 with a lesser weight, the amplitude of the summate voltage becomes significantly smaller due to its faster decay. Throughout the process, the summate potential cannot reach the threshold voltage of the memristor, resulting in a minimum output corresponding to a lower voltage of 0, as illustrated in Figure 5f. In contrast, when signal2 arrives at S2 with a small weight before signal1 arrives at S1 with a greater weight, the amplitude of the summate voltage exceeds the threshold voltage. As a result, an output pulse with a high voltage of 1 is generated as shown in Figure 5i. A similar output pulse was also obtained from other synapses such as S1 and S3, S4 and S5, S4 and S6, and so on, when their weights are different to each other. Consequently, an audio signal-stimulated SNN hardware based on multilayered HfO_x_/TiO_y_ neurons and synapses was successfully realized.

## 4. Conclusions

In summary, we successfully designed and fabricated multilayered HfO_x_/TiO_y_ memristor crossbar arrays that exhibited remarkable synapse-like resistive switching characteristics and neuron-like integrate-and-fire behavior. It is found that tunable volatile and nonvolatile switching in the multilayered HfO_x_/TiO_y_ memristor is induced by the size-controlled atomic oxygen vacancy pathway, which depends on the atomic sublayer in the multilayered structure. The tunable threshold switching characteristic can be successfully employed to mimic the integrate-and-fire function of biological neurons. Based on the stable performance of the multilayered HfO_x_/TiO_y_ neurons and synapses, we designed an SNN hardware architecture for processing audio signals, which provides a base for the recognition of audio signals through the function of integration and firing. Our implementation of synaptic functionalities by the hardware of SNNs paves the way for novel neuromorphic computing paradigms in the AI era.

## Figures and Tables

**Figure 1 nanomaterials-14-01412-f001:**
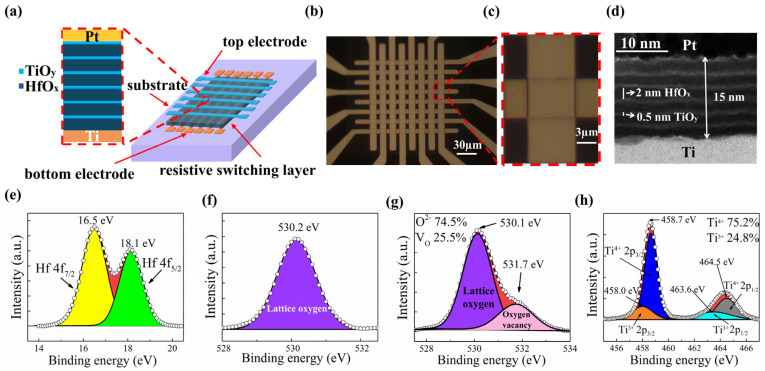
(**a**) An illustration showcasing the structure of the multilayered HfO_x_/TiO_y_ memristor crossbar array; (**b**,**c**) the optical microscope image depicting the multilayered HfO_x_/TiO_y_ crossbar and a detailed picture of the intersection node; (**d**) a high-resolution TEM photograph of the HfO_x_/TiO_y_ multilayers; (**e**,**f**) the XPS spectra of Hf 4f and O 1s within the HfO_x_ sublayer, where yellow represents Hf 4f_7/2_, the green peak represents Hf 4f_5/2_, and the purple peak stands for lattice oxygen; (**g**,**h**) the XPS spectra of Ti 2p and O 1s in the TiO_y_ sublayer, in which the purple peak and pink one represents lattice oxygen and oxygen while the blue, orange, gray and cyan peaks stand for Ti^4+^ 2p_3/2_, Ti^3+^ 2p_3/2_, Ti^4+^ 2p_1/2_ and Ti^3+^ 2p_1/2_, respectively.

**Figure 2 nanomaterials-14-01412-f002:**
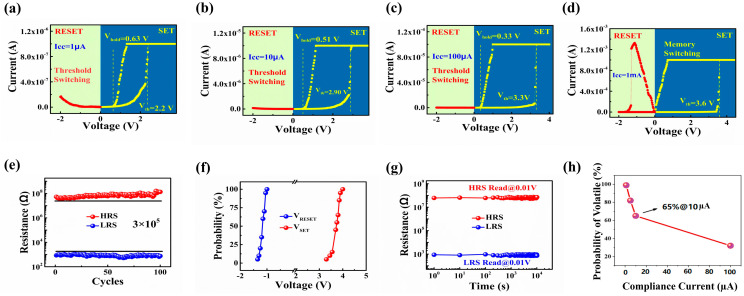
The threshold switching behavior of the multilayered HfO_x_/TiO_y_ memristor was observed at compliance currents of (**a**) 1 µA, (**b**) 10 µA, and (**c**) 100 µA, respectively; (**d**) the memory switching properties of the multilayered HfO_x_/TiO_y_ memristor in the set process with an Icc of 1 mA; (**e**) the endurance characteristic of the multilayered HfO_x_/TiO_y_ memristor after 300 cycles under DC sweeping mode; (**f**) the distribution of variations in Vset, Vreset, and LRS/HRS values among different devices was analyzed using data collected from 8 memristor devices; (**g**) the retention properties of the multilayered HfO_x_/TiO_y_ memristor device under ambient conditions; (**h**) the statistical probability of threshold switching from the multilayered HfO_x_/TiO_y_ memristor device with a different Icc.

**Figure 3 nanomaterials-14-01412-f003:**
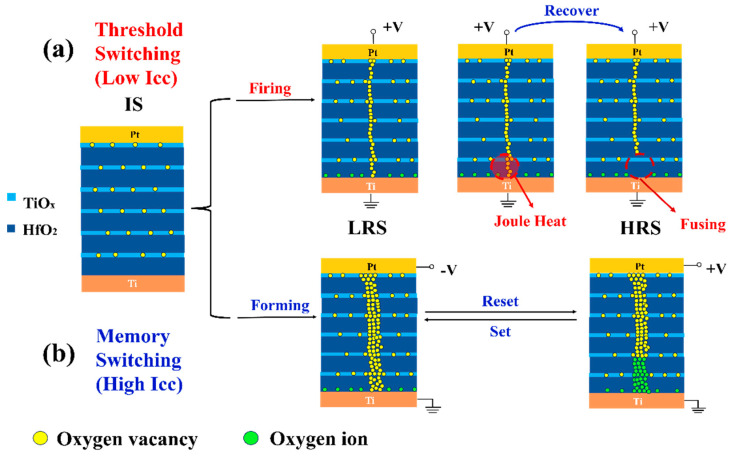
(**a**,**b**) A schematic diagram of the switching pathway model related to threshold switching and memory switching of the multilayered HfO_x_/TiO_y_ memristor under different Iccs.

**Figure 4 nanomaterials-14-01412-f004:**
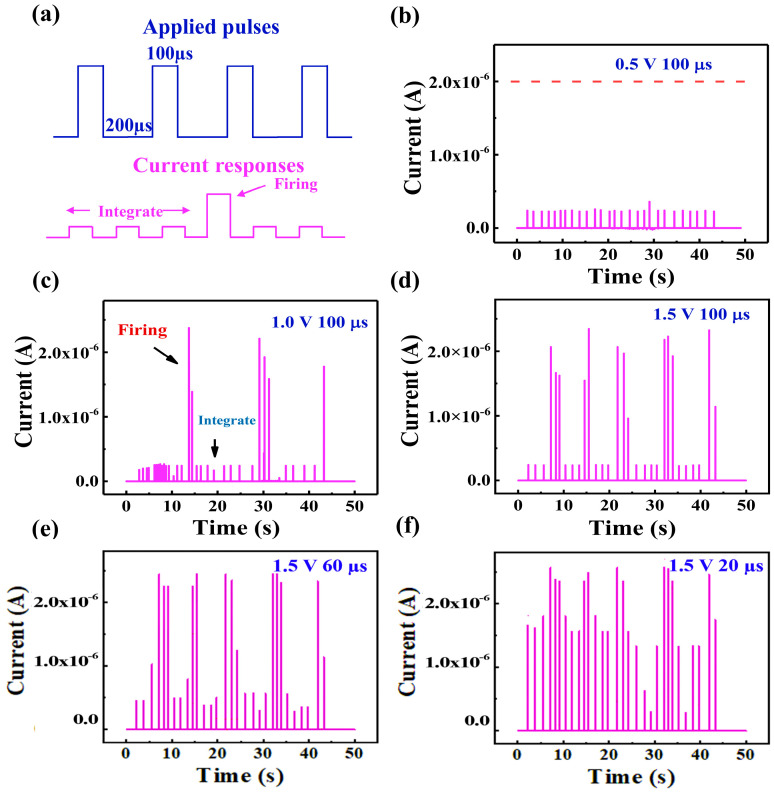
(**a**) The method and origin of the integrate-and-fire function of the multilayered HfOx/TiOy neuron tests. (**b**–**d**) Output outcomes are observed at amplitudes of 0.5 V, 1 V, and 1.5 V, respectively, with no firing detected at an amplitude of 0.5 V. The firing is initiated at an amplitude of 1 V. When the voltage pulse’s amplitude reaches 1.5 V, there is a corresponding increase in the firing frequency. (**d**–**f**) Maintaining the input pulse amplitude at 1.5 V, the firing frequency rises as the pulse width decreases from 100 µs to 60 µs and 20 µs.

**Figure 5 nanomaterials-14-01412-f005:**
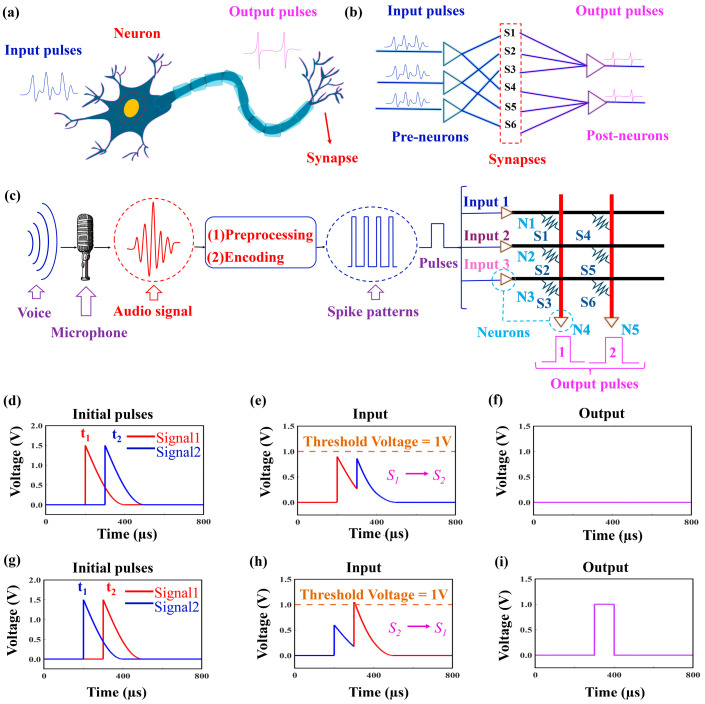
(**a**) The diagrammatic representation of a biological neuron; (**b**) an illustrative diagram outlining the realization of SNN hardware, comprising neurons and synapses; (**c**) an audio signal processing system based on an SNN hardware model; (**d**,**g**) input signal pulses; (**e**,**h**) the output of pre-neurons represented by the potential of the neurons; (**f**,**i**) the output of post-neurons represented by voltages of 0 and 1.

## Data Availability

The original contributions presented in the study are included in the article, further inquiries can be directed to the corresponding author.

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
