# Peer review of "Audio Signal-Stimulated Multilayered HfOx/TiOy Spiking Neuron Network for Neuromorphic Computing"

_nanomaterials, 2024, doi:10.3390/nano14171412_

Round 1

Reviewer 1 Report

Comments and Suggestions for Authors

Comments on the Quality of English Language

It's fine.

Reviewer 2 Report

Comments and Suggestions for Authors

In this manuscript, the authors report a significant advance in neuromorphic computing through the creation of multilayered HfOx/TiOy memristor crossbar arrays. These devices exhibit switching behavior that mimics synapses and neurons, with the ability to control this switching by adjusting atomic pathways—a key innovation. Overall, this work makes a strong contribution, providing a novel approach to integrating synaptic functions into hardware, which is crucial for the future of AI. The successful application of these concepts in spiking neural networks (SNNs) for audio recognition suggests promising real-world potential. However, there are a few suggestions for further improvement:

1) What specific advantages does the multilayered HfOx/TiOy memristor structure offer over single-layer memristors in terms of synapse-like resistive switching and neuron-like behavior? The authors should provide a comparison, either from their experiments or from previously reported results in the literature.

2) How does the period of the multilayered HfOx/TiOy films affect the device properties? Additionally, what about the thicknesses of the sublayers? Have the authors optimized the thickness used here, and are there potential improvements?

3) Given that the films were deposited at high temperatures, what is the roughness of the films, and does this lead to a large leakage current in the devices?

Reviewer 3 Report

Comments and Suggestions for Authors

The manuscript is an interesting report that artificial neurons and synapses based on multilayered HfOx/TiOy memristor crossbar arrays can be used for SNN training of audio signals. I would recommend publishing after addressing a few issues:

1.      The authors should add qualitative analysis in the abstract.

2.      Can authors clearly explain how they made layer-by-layer growth of HfOx and TiOx, as shown in Figure 1d?

3.      Also, the scale does not look compatible in Figure 1d. Can authors explicitly mention/mark the Ti, TiOx, HfOx, and Pt in Figure 1d?

4.      The scale bar is missing in Figure 1b.

5.      It is a point of curios that the thickness of TiOx (0.5 nm) is controlled by ALD. Explain?

6.      What is the thickness of Ti and Pt electrodes?

7.      Please explain in detail why the Vset becomes large and the compliance current is increased.

8.      It is better to add recent articles on oxide-based memristors.

a.       https://doi.org/10.1016/j.jallcom.2024.175103

b.      https://doi.org/10.1002/adfm.202300343

c.       https://doi.org/10.1016/j.mseb.2023.116755

9.      How high is Icc linked with the memory window?

10.   What is the motivation for keeping TiOx thickness (0.5 nm)? Did the author have some data with TiOx thickness (1 or 2 nm)?

Comments on the Quality of English Language

 Minor editing of English language required.

Round 2

Reviewer 1 Report

Comments and Suggestions for Authors

The manuscript has been improved after revision and its quality meets the journal's standard. The reviewer recommends publication as it is.  

Reviewer 3 Report

Comments and Suggestions for Authors

The authors responded to almost all the questions raised in the report. I would like to request that the editor accept this manuscript for publication.